# Doubly residual transitions for deep variational state-space models

## Abstract

Sequential data modeling often relies on capturing underlying dynamics through Variational State-Space Models (VRSSMs), yet the architecture of transition functions in these models remains underexplored. Here we investigate highway layers as latent transitions in VRSSMs, leveraging their trainable gating mechanisms that allow flexible combination of raw and transformed representations. Through extensive empirical evaluation across multiple datasets, we demonstrate that highway transitions consistently outperform standard multi-layer perceptron (MLP) baselines. Our results show that highway-based VRSSMs achieve better validation performance while demonstrating enhanced robustness to hyper-parameter choices. The findings highlight how established neural network techniques can significantly impact probabilistic sequential modeling when applied in new contexts. We recommend that practitioners incorporate highway connections in their modeling toolbox for VRSSMs, as they provide a simple yet effective architectural enhancement for capturing temporal dependencies in sequential data.

## 1 Introduction

The machine learning community has long recognized that successful model training often hinges on architectural innovations and practical heuristics that address challenges such as optimization instability and poor convergence. While many of these techniques, like residual connections (He et al., 2015) and highway layers (Srivastava et al., 2015), have been extensively studied and celebrated for their role in advancing the pursuit of deeper neural network models by mitigating the vanishing gradient problem (Hochreiter, 1991; He et al., 2016; Zilly et al., 2017), often, their application in specific contexts remains underexplored or their impact undocumented. For example, the potential of these mechanisms to improve latent transitions in sequential probabilistic models has yet to be fully understood.

We address this gap and investigate the impact of highway transitions in deep VRSSMs. Deep VRSSMs are designed to identify dynamics that underpin sequential data (Karl et al., 2017). Despite this focus, investigations into the impact of the transition architecture have been secondary. This paper contributes to closing this gap by exploring the advantages of highway layers as latent transitions in VRSSMs. Highway layers, with their trainable gating mechanisms, allow a flexible combination of raw and transformed representations, offering a promising approach to enhance sequential data modeling with VRSSMs. We show that VRSSMs with highway transitions improve over those with baseline, MLP transitions by balancing latent encoding and data reconstruction more effectively.

By building on the foundational work of residual and highway connections, this paper positions highway layers as a powerful tool for advancing dynamical systems modeling with VRSSMs. Our findings highlight the importance of revisiting established techniques in new contexts, underscoring their potential to inspire innovation at the intersection of deep learning and probabilistic modeling.

## 2 Background

### 2.1 State-space modeling for sequential data

State-space models (SSMs) provide a principled framework for modeling sequential observations by introducing latent variables that capture underlying temporal dependencies (Särkkä, 2013). Instead of modeling observations directly, SSMs assume that the observed data $\boldsymbol{x}_t$ is generated from an evolving latent state $\boldsymbol{z}_t$, which follows a structured, probabilistic transition process:

$$p\left(\mathbf{z}_t \mid \mathbf{z}_{t-1}\right), \quad \text{(transition model)} \tag{1}$$

$$p\left(\mathbf{x}_t \mid \mathbf{z}_t\right). \quad \text{(observation model)} \tag{2}$$

SSMs naturally arise in systems where observations are noisy or incomplete representations of an underlying process. For example in robotic motion, the latent state may encode position and velocity, evolving according to physical laws, while noisy camera images serve as observations.

Central to state-space modeling is the Markov assumption, i.e. the current latent state $\boldsymbol{z}_t$ depends only on the previous state $\boldsymbol{z}_{t-1}$, and each observation $\boldsymbol{x}_t$ is conditionally independent of past observations given $\boldsymbol{z}_t$. This assumption allows for efficient inference while maintaining a rich representation of sequential dependencies. For a sequence of observations $\boldsymbol{x}_{0:T} = \{\boldsymbol{x}_0, ..., \boldsymbol{x}_T\}$, the joint distribution can be factorized as:

$$p\left(\mathbf{x}_{0:T}, \mathbf{z}_{0:T}\right) = p\left(\mathbf{z}_0\right) \prod_{t=1}^{T} p\left(\mathbf{z}_t \mid \mathbf{z}_{t-1}\right) \prod_{t=1}^{T} p\left(\mathbf{x}_t \mid \mathbf{z}_t\right), \tag{3}$$

where $p\left(\mathbf{z}_0\right)$ represents the initial state distribution, and marginalizing out the latent states yields the observation likelihood.

In this paper, we use SSMs to structure observational data, leveraging their ability to capture temporal dependencies and account for noise. Complex transition dynamics in real-world systems often require flexible transition models to capture intricate relationships across time. When applied to long time series, this naturally leads to deep models along the time dimension, where information must be processed and propagated across many steps. Such depth necessitates neural architectures that incorporate mechanisms for controlling information flow to prevent degradation of long-term dependencies, as well as stable and efficient training (Pascanu et al., 2013). In particular, gating mechanisms have emerged as a powerful solution, dynamically regulating memory updates and suppressing irrelevant information.

### 2.2 Gating mechanisms in deep learning models

Gating mechanisms selectively regulate information flow in neural networks, ensuring efficient memory retention and transformation, and enhance gradient propagation and adaptive computation (He et al., 2015; 2016; Vaswani et al., 2023; Liu et al., 2021). A generic gating function $g : \mathbb{R}^d \to [0, 1]^d$ is given by:

$$g\left(\boldsymbol{x}\right) = \sigma\left(f_\theta\left(\boldsymbol{x}\right)\right), \tag{4}$$

where $\theta$ are learnable parameters, and $\sigma\left(\cdot\right)$ is typically a sigmoid activation function.

Gating mechanisms are crucial components in various successful neural architectures, including, transformers (Vaswani et al., 2023), spatial gating units (Liu et al., 2021), gated recurrent networks (Hochreiter & Schmidhuber, 1997; Cho et al., 2014), and mixture-of-experts (MoE)-layers (Shazeer et al., 2017) each adapting gating principles differently. Highway layers modulate the information flow via:

$$\boldsymbol{x}_{t+1} := \text{highway}_\theta\left(\boldsymbol{x}_t\right) := g_\theta\left(\boldsymbol{x}_t\right) \odot h_\theta\left(\boldsymbol{x}_t\right) + \left(1 - g_\theta\left(\boldsymbol{x}_t\right)\right) \odot \boldsymbol{x}_t, \tag{5}$$

where $h_\theta\left(\boldsymbol{x}\right)$ represents a transformation of $\boldsymbol{x}$, and $\odot$ is the element-wise Hadamard product. Used as recurrent units, highway layers dynamically balance memory retention and transformation across time, providing a structured mechanism information flow particularly suited for sequence modeling (Zilly et al., 2017). In this work, we extend this concept to deep state-space models, leveraging highway layers as transition functions to enhance latent state dynamics and improve sequence modeling.

### 2.3 Amortized variational inference

Amortized variational inference Kingma & Welling (2022) is a scalable approach to approximate Bayesian inference for complex latent variable models. The goal is to fit the parameters of a latent variable model $p_\theta(\mathbf{x}, \mathbf{z}) = p_\theta(\mathbf{x}|\mathbf{z}) p_\theta(\mathbf{z})$ such that its marginal likelihood $p_\theta(\mathbf{x})$ aligns with a target data distribution $p(\mathbf{x})$. This is achieved by maximizing the log marginal likelihood:

$$\arg\max_\theta \mathbb{E}_{\mathbf{x} \sim p(\mathbf{x})} \left[ \log p_\theta(\mathbf{x}) \right]. \tag{6}$$

Here, $\mathbf{x}$ represents the observed data. However, computing $\log p_\theta(\mathbf{x})$ involves the marginalization over the latent variables $\mathbf{z}$:

$$p_\theta(\mathbf{x}) = \log \int p_\theta(\mathbf{x}, \mathbf{z}) \, d\mathbf{z}, \tag{7}$$

which is often intractable due to the high-dimensional integral. To address this challenge, variational inference introduces a surrogate posterior $q_\phi(\mathbf{z}|\mathbf{x})$, parameterized by $\phi$, to approximate the true posterior $p_\theta(\mathbf{z}|\mathbf{x})$. The evidence lower bound (ELBO) provides a tractable alternative to the log marginal likelihood:

$$\log p_\theta(\mathbf{x}) \geq \mathbb{E}_{q_\phi(\mathbf{z}|\mathbf{x})} \left[ \log p_\theta(x|z) \right] - \mathrm{KL} \left[ q_\phi(\mathbf{z}|\mathbf{x}) | p_\theta(\mathbf{z}) \right]. \tag{8}$$

Maximizing the ELBO aligns the latent variable model $p_\theta(\mathbf{x}, \mathbf{z})$ with the data distribution $p(\mathbf{x})$ by:

- Encouraging the latent representations $\mathbf{z}$ under $q_\phi$ to reconstruct the observed data $\mathbf{x}$ accurately via $\mathbb{E}_{q_\phi(z|x)} \left[ \log p_\theta(x|z) \right]$.

- Minimizing the Kullback–Leibler (KL)-divergence between the approximate posterior $q_\phi(\mathbf{z}|\mathbf{x})$ and the prior $p_\theta(\mathbf{z})$, thereby regularizing the latent space.

This optimization reframes the Bayesian inference problem of posterior estimation as a parameter optimization problem. Notably, the ELBO eliminates the need to compute $\log p_\theta(\mathbf{x})$ directly, avoiding the intractable integral over $\mathbf{z}$.

Amortized variational inference further extends classical variational methods by sharing the parameters $\phi$ of $q_\phi(\mathbf{z}|\mathbf{x})$ across data points through a neural network. Instead of separately optimizing a posterior distribution for each sample, a global inference model learns to map $\mathbf{x}$ to $q_\phi(\mathbf{z}|\mathbf{x})$, greatly improving scalability for large datasets.

## 3 Method

For our study, we use residual SSMs (Karl et al., 2017; Sölch, 2021), which constitute a particular class of SSMs. Residual SSMs construct the transition distribution from a deterministic component plus a component-wise scaled residual. Both components are implemented as feed-forward neural networks (FFNs) parametrized by learnable parameters $\theta$:

$$\mathbf{z}_t = FFN_{\theta_{det}}(\mathbf{z}_{t-1}, \boldsymbol{u}_{t-1}) + FFN_{\theta_{res}}(\mathbf{z}_{t-1}, \boldsymbol{u}_{t-1}) \odot \varepsilon_t, \quad \varepsilon_t \sim \mathcal{D}_\varepsilon, \tag{9}$$

where $\theta_{det}, \theta_{res} \subset \theta$, $\boldsymbol{u}_t$ are additional conditions, e.g. control signals, and the residual distribution $\mathcal{D}_\varepsilon$ is assumed to be zero-centered.

### 3.1 Deep state-space models with highway transitions

On the residual nature of residual SSMs, we add a second residual layer, namely highway transitions. While the former induces SSM structure in the variational posterior by enabling the reuse of the deterministic component in the inference network, residual connections in neural network layers serve the purpose of explicit identity propagation. We use highway layers as the deterministic component of the transition function:

$$FFN_{\theta_{det}}(\mathbf{z}_{t-1}, \boldsymbol{u}_{t-1}) = \mathrm{highway}_{\theta_{det}}([\mathbf{z}_{t-1}, \boldsymbol{u}_{t-1}]), \tag{10}$$

where $[\cdot, \cdot]$ denotes the concatenation of the bracket content.

For learning SSM, we use amortized variational inference, which is particularly well-suited for learning the underlying dynamics in sequential latent variable models. By leveraging both, recognition and reconstruction models, we can incentivize structuring the latent space to align with state-space modeling assumptions. This is crucial for accurate long-term predictions (Karl et al., 2017).

### 3.2 Amortized variational inference in state-space models

Similar to the static case, the ELBO optimization makes inference in SSMs tractable by introducing an approximate posterior $q_\phi\left(\mathbf{z}_{0:T}|\mathbf{x}_{0:T}\right)$, enabling the ELBO formulation. We chose the approximate posterior model $q_\phi$ that decomposes like an SSM, and observe that also the sequential ELBO decomposes over time (Bayer et al., 2021; Sölch, 2021):

$$\text{ELBO} = \sum_{t=0}^{T} \mathbb{E}_{q_\phi(\mathbf{z}_t|\mathbf{x}_{0:T})}\left[\log p_\theta\left(\mathbf{x}_t|\mathbf{z}_t\right)\right] \tag{11}$$

$$- \text{KL}\left[q_\phi\left(\mathbf{z}_0|\mathbf{x}_{0:T}\right)|p_\theta\left(\mathbf{z}_0\right)\right] - \sum_{t=1}^{T} E_{q_\phi(\mathbf{z}_t|\mathbf{x}_{0:T})}\left[\text{KL}\left[q_\phi\left(\mathbf{z}_t|\mathbf{z}_{t-1}, \mathbf{x}_{t:T}\right)|p_\theta\left(\mathbf{z}_t|\mathbf{z}_{t-1}\right)\right]\right]. \tag{12}$$

We refer to the sequential ELBO's terms as reconstruction error, initial prior divergence, and expected transition prior divergence, respectively. The combination of SSMs and posterior approximation via the ELBO creates the VRSSM approach. We benefit from the choice of approximate posterior factorization as the inference model can reuse the deterministic component and needs to infer only the residual. This incentivizes SSM-structure on the latent space via both, inference and reconstruction model. For modeling details, we refer to Karl et al. (2017) and Sölch (2021).

## 4 Experiments

We investigate the effect of highway layers when used as the deterministic component $FNN_{\theta_{det}}$ in the transition of deep variational state-space models (compare Equation 9). Therefore, we limit our discussion to features of the deterministic component in the remainder of the script.

Our implementation of the highway layer is based on an MLP with an output of size twice the number of latent variables. The first half of the output is fed into the activation function $\sigma$ to generate the weighting coefficients, while the second half forms the output of the sub-network $h_\theta$ of Equation 5. We empirically compare the highway transition to an MLP transition. Our model implementation is the same as in Bayer et al. (2021). Subsection B.1 lists the architecture details and parameters used in the experiments.

Our experimental framework conducts a systematic comparison between highway networks and standard MLP transitions. To establish practically relevant insights, we evaluate 500 sampled configurations from a search space that covers commonly tuned hyperparameters (detailed in Subsection B.4), training paired MLP and highway-enabled VRSSMs models for each configuration. The evaluation includes diverse systems: the pendulum, with two degrees-of-freedom (DoF) and deterministic dynamics, serves as a baseline; the hopper system features higher DoF and more complex dynamics; and sequential MNIST presents a case with stochastic dynamics. For data acquisition and preprocessing checkout Appendix C. All models undergo identical training procedures, allowing us to isolate architectural effects from other confounding factors.

### 4.1 Results

We evaluate the models on the ELBO. Figure 1 depicts the advantage of VRSSMs with highway-transition against their MLP-transition counterpart. It compares the performance of the pendulum transition variants by ranked validation loss. For the pendulum in Figure 1(a), we observe that highway-transitions give only a small edge among the top-ranking models. Towards larger ranks the performance gap widens in favor of highway transitions. Since machine learning practitioners commonly conduct a hyperparameter search

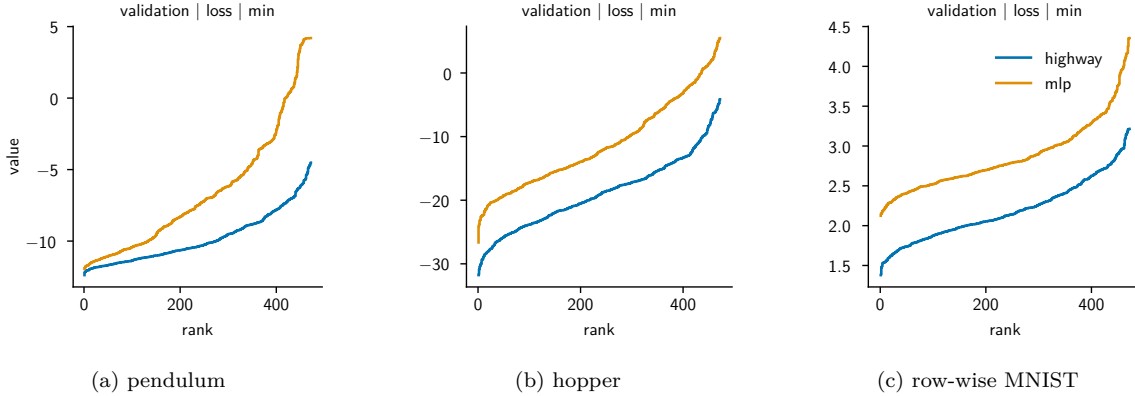

Figure 1: The plots overlay the ranked runs of highway and mlp transition. They perform ranking according to the minimum validation loss value recorded throughout training.

to determine the best model configuration, these results do not indicate a practical advantage of highway-transitions. However, they make it more likely to find a good model given an arbitrary hyperparameter configuration.

On the other two datasets in Figure 1(b) and Figure 1(c) respectively, we observe a performance gap across all ranks. Firstly, this strengthens the observation that highway-transitions can be tuned more easily. Secondly, it gives evidence, that highway-transitions yield better VRSSMs.

For qualitative evaluation of predictive capabilities, we initialize model predictions from an approximate filtering distribution. This methodological choice intentionally excludes future observations to isolate visualization of the generative model's stochasticity. Although the inference model estimates a smoothing posterior, we strategically employ it to construct a filtering approximation (detailed in Subsection D.2), leveraging the learned representations while maintaining temporal causality. Predictions propagate from the final filtered state, with faithful reconstructions in Figure 2 and Figure 3 confirming the filter's effectiveness across both architectures. Example predictions initially follow the ground truth observations closely, but deteriorate over time. This gradual divergence is an expected consequence of the stochastic nature of the dynamics model, as uncertainty accumulates with each time step. The depicted rollouts represent individual realizations from a distribution of possible futures modeled by the system. Interestingly, Figure 3 demonstrates that predictions maintain fidelity to ground truth observations even beyond the training horizon, suggesting robust capture of long-term dynamics.

On the MNIST dataset, the inference-model-based filtering approach from Subsection D.2 does not yield faithful posterior estimates, and, hence, does not allow for a qualitative evaluation of the predictive model. We add examples in Figure 5. Note, that the construction of an alternative filter, e.g. a particle filter, from the learned model is possible.

As an alternative evaluation, we present reconstructions from the model's inferred initial state in Figure 4. We call them pseudo-reconstructions. Subsection D.1 contains details on their acquisition. Pseudo-reconstructions cannot visualize the stochasticity of the rollout well, since the estimate of the initial state already contains information on the entire time series. However, the examples demonstrate a well-behaving deterministic component of the transition model.

## 5 Related Work

### 5.1 Foundations of deep sequential latent variable models

The evolution of deep sequential models has seen significant advancements in both deterministic and stochastic frameworks. Long short-term memory (LSTM) networks (Hochreiter & Schmidhuber, 1997) and gated

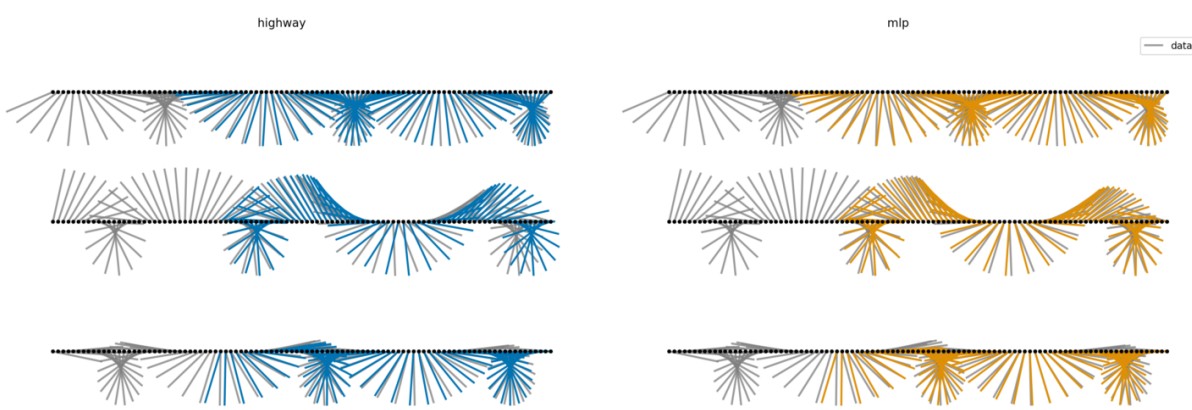

Figure 2: The plots depict predictive posteriors with a prefix length of 33 time steps for three example trajectories (top to bottom) from the validation dataset. The total trajectory length is 100 time steps. Details on the acquisition of posterior predictive samples are in Subsection D.2.

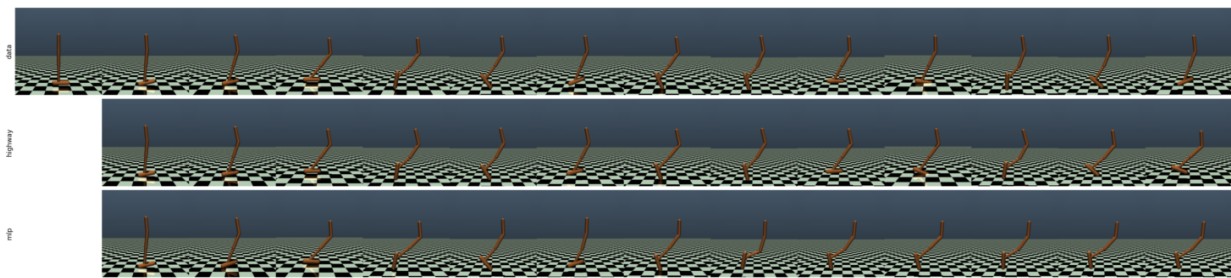

Figure 3: The plots depict predictive posteriors with a prefix length of ten filtering time steps. The total trajectory length is 500 time steps of which we visualize 15 sampled at regular intervals. Note that, we use custom code in Algorithm 1 to render hopper observations trajectories.

recurrent units (GRUs) (Cho et al., 2014) established foundational gating mechanisms for gradient-stable temporal modeling. Pascanu et al. (2014)'s work on "How to Construct Deep Recurrent Neural Networks" provided crucial insights into deepening these architectures, exploring various strategies for enhancing expressivity and gradient flow.

Zilly et al. (2017)'s recurrent highway network (RHN) further extended these concepts by introducing residual highway connections to deepen transitions, enhancing information flow in deterministic settings. These developments in deterministic models paved the way for more complex stochastic approaches ().

### 5.2 Architectural innovations in deep neural networks

The challenge of training very deep networks was addressed by Highway Networks (Srivastava et al., 2015) and residual networks (ResNets) (He et al., 2015). Highway Networks employ LSTM-inspired gating mechanisms to regulate information flow across layers, while ResNets utilize identity skip connections to facilitate gradient propagation(He et al., 2016). Both architectures mitigate the vanishing gradient problem (Hochreiter, 1991), enabling the training of networks with hundreds of layers and revolutionizing deep learning across various domains ().

### 5.3 Probabilistic state-space models

The integration of variational inference with SSMs led to the development of deep VRSSMs. Karl et al. (2017)'s deep variational Bayes filters (DVBFs) stands out for its approach to system identification, employing neural networks to parameterize nonlinear transitions while maintaining latent Markovian structure. DVBF

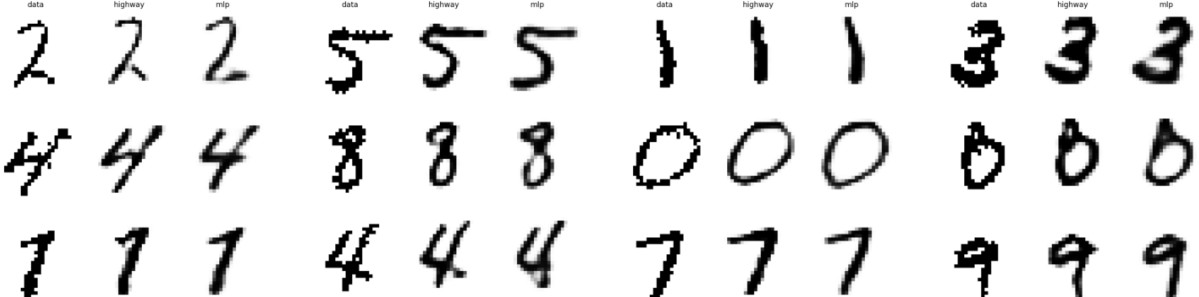

Figure 4: The plots depict pseudo reconstructions for ten examples from the validation data set.

demonstrates the ability to handle highly nonlinear input data with temporal and spatial dependencies without domain-specific knowledge.

In contrast, Hafner et al. (2019)'s recurrent state-space modelss (RSSMs) adopt a shallow highway-inspired GRU core for deterministic propagation within a broader stochastic framework. While RSSMs focus on capturing dynamics through a combination of categorical latent representations and deterministic transitions, they do not enforce the same strict latent Markovian structure as DVBFs. Similarly, Lee et al. (2020)'s stochastic latent actor critic (SLAC) combines hierarchical latent variables with reinforcement learning but lacks the robust model identification features inherent in DVBF.

Recent innovations by Gu et al., including structured state space (S4) (Gu et al., 2022) and Mamba (Gu & Dao, 2024) models, have achieved linear-time sequence modeling with long-range dependencies through parallel time-axis computations and high-order polynomial projection operators (HiPPO) initialization. Mamba enhances S4 with input-dependent gating mechanisms ("selectivity"), though both remain limited to deterministic, linear state transitions.

### 5.4 Model identification and highway transitions

A key challenge in state-space modeling is balancing transition expressivity with model identifiability. DVBF addresses this through structured transitions and inference reuse, sharing parameters between recognition and generative networks to enforce state-space structure. This approach facilitates learning interpretable latent states and enables efficient unsupervised learning of state-space models.

While gating mechanisms and highway networks have proven successful in deterministic models, their impact on stochastic state-space models remains underexplored. The specific benefits of these architectural choices on model identification and expressiveness in probabilistic settings warrant further investigation.

Our work aims to bridge this gap by incorporating highway connections into the DVBF framework, providing an empirical analysis of highway benefits in this context. This contribution advances the design of effective, probabilistic, and interpretable latent variable models for sequential data.

## 6 Conclusion

We have presented evidence that VRSSMs benefit from highway transitions. The benefit holds across systems with varying complexity of their dynamics, and both, deterministic, and stochastic dynamics. Further, it holds robustly across a range of reasonable and useful hyperparameter sets. Our findings align with previous ones on deep neural networks and recurrent highway networks. Hence, we add sequential, variational models to the field of application of highway connections. We can strongly recommend practitioners consider having highway connections in their active toolbox.

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

# A  Appendix

# B  Model architecture and parametrization

## B.1  VRSSM architecture

Our concrete implementation follows the fully conditioned variant in Bayer et al. (2021). Across datasets, the Vrssm, the Vrssm-Machine, and the Vrssm-Learner architecture share the following structure and hyperparameters.

Table 1: VRSSM Architecture: Components and Parameters

| Component | Realization | Description |
|---|---|---|
| **number of latent variables** | compare Subsection B.4 | The dimensionality of the latent space per time step. |
| **prior initial** | normalizing flow | Parametric prior distribution of the initial state. |
| normalizing flow | real NVP (Dinh et al., 2017) | Base distribution of the real NVP. The parameters of the base distribution are part of the optimized parameter set. |
| base | multi-variate Gaussian | Base distribution with diagonal covariance. |
| number of flows | 12 | Number of stacked flows. |
| coupling layer | affine | Invertible transformations used in normalizing flows. |
| shift and scale functions | MLP | The parametric functions determine the shift and scale values for the affine transformation. |
| **infer initial** | conditional Gaussian distribution | Conditional parametric posterior distribution of the initial state. |
| conditions_to_pars | MLP | Neural network parameterizing the distribution parameters. |
| location | — | Location parameters are output by conditions_to_pars. |
| scale_diag | — | Scale parameters are output by conditions_to_pars. |
| **prior disturbance** | zero-centered multi-variate Gaussian | Prior distribution over the stochastic part of the transition with a diagonal covariance matrix. |
| **infer disturbance** | conditional Gaussian distribution | Conditional parametric posterior distribution over the stochastic component of the transition distribution. |
| residual scalar | MLP | — |
| **feature extract** | RNN | — |
| **number of features** | compare Subsection B.4 | Number of output features summarizing succeeding observations-condition trajectories. |
| **transition** | MLP, highway | Deterministic component of the latent dynamics. |

| emission | conditional Gaussian distribution | Conditional parametric posterior distribution of the initial state. |
|---|---|---|
| conditions_to_pars | parallel [MLP, scales] | Parameterizes the neural network whose output is the distribution parameters. |
| location | MLP | Location parameters are the output of the MLP. |
| scale_diag | scales | Scale parameters are the output of the scales layer. |
| **MLP** | — | Multi-layer Perceptron. |
| n_hidden | compare Subsection B.4 | — |
| activation | softsign | — |
| n_layers | 1 | — |
| use_layer_norm | true | — |
| **RNN** | — | Recurrent Neural Network. |
| n_hidden | compare Subsection B.4 | — |
| activation | softsign | — |
| n_layers | 1 | — |
| cell | GRU | — |
| backwards | true | Flag to flip the input sequence. |

## B.2   Machine

Table 2 lists the machine parameters used across experiments.

Table 2: Machine's parameters.

| Component | Realization | Description |
|---|---|---|
| optimizer | adam (Kingma & Ba, 2017) | |
| step size | compare Subsection B.4 | Learning rate of the optimizer. |
| use initial kl | false | Since we do not include KL divergence of the initial state in the sequential ELBO loss, the parameters of the prior of the initial state distribution are not optimized. |

## B.3   Learner

Table 3 lists the learner parameters used across experiments.

Table 3: Learner's parameters.

| Component | Realization | Description |
|---|---|---|
| batch size | compare Subsection B.4 | |
| max iter | 30000 | Number of training iterations. |
| report interval | 500 | Report interval in the number of training iterations. |

## B.4 Hyperparameter search domain

For our empirical comparison, we design a search space over commonly tuned hyperparameters in VRSSM training, namely, the optimizer step size, the MLP's number of hidden units, recurrent neural network (RNN)'s number of hidden units, its number of output features, the number of latent variables of the SSM, and the batch size. For a detailed specification of the search domain check out Table 4. We use the same search domain for all data sets. From this space, we draw 500 samples. For each set, we train one highway-transition and one MLP-transition VRSSM. This allows the evaluation of the aligned models in e.g. Figure 6.

Table 4: Pendulum parameters.

| Parameter | Space | Description |
|---|---|---|
| number of latent states | $\{2, 3, \ldots, 256\}$ | from discrete set |
| number of features | $\{2, 3, \ldots, 256\}$ | from discrete set |
| batch size | $\{64, 128, 256, 512\}$ | from discrete set |
| MLP - Number of hidden units | $\{2, 3, \ldots, 256\}$ | from discrete set |
| RNN - number of hidden units | $\{2, 3, \ldots, 256\}$ | from discrete set |
| optimizer's step size | $[0.001, 0.01]$ | from continuous, logarithmic range |

## C Datasets

### C.1 Pendulum

We base our dataset on the random v1 version of the pendulum dataset from the gymnasium suite (Fu et al., 2021; Towers et al., 2024). For the specification and interpretation of the observation and action spaces check the original publication. We extract action-observation trajectories by cutting all possible windows of length 100 time steps. We keep all dimensions of both, observation and action space. Splitting into training-validation-test partition yields sets of size 4500, 1350, and 3150 trajectories, respectively. We standardize all sets by the per-dimension mean and standard deviation of the training data set.

### C.2 Hopper

We base our dataset on the expert v2 version of the hopper dataset from the gymnasium suite (Fu et al., 2021; Towers et al., 2024). For the specification and interpretation of the observation and action spaces check the original publication We extract action-observation trajectories by cutting all possible windows of length 28 time steps. We keep all dimensions of both, observation and action space. Splitting into training-

validation-test partition yields sets of size 49821, 14946, and 34876 trajectories, respectively. We standardize all sets by the per-dimension mean and standard deviation of the training data set.

### C.3 Row-wise MNIST

We take the original MNIST dataset (LeCun, 1998) and create a binary and sequential version. First, we binarize by sampling from a Bernoulli distribution with a rate proportional to the corresponding pixel intensity. Secondly, we treat the sequence of rows in the image plane as a temporal sequence of observations. Our sequential Modified National Institute of Standards and Technology (MNIST) dataset does not contain actions. As multiple numbers share similar initial rows, e.g. 3, 8, 9, or 0, this yields trajectories with stochastic dynamics. This derivation was previously used in Bayer et al. (2021). We use a Bernoulli emission distribution to model the binary nature of the data.

## D Results

### D.1 Pseudo reconstructions

We present qualitative results in the form of pseudo reconstructions in Figure 4. We obtain pseudo-reconstructions via ancestral sampling from the initial state's posterior. In detail, we draw a sample from the initial state's posterior. We roll out trajectories from this sample using the learned VRSSM.

$$p\left(\overset{\rightrightarrows}{\mathbf{o}}_t|\boldsymbol{o}_{0:T},\boldsymbol{u}_{1:T}\right) = \int q_\theta\left(\mathbf{z}_0|\boldsymbol{o}_{0:T}\boldsymbol{u}_{1:T}\right)\left[\prod_{\tau=1}^{t} p_\theta\left(\mathbf{z}_\tau|\mathbf{z}_{\tau-1},\boldsymbol{u}_{\tau-1}\right)\right] p_\theta\left(\overset{\rightrightarrows}{\mathbf{o}}_t|\mathbf{z}_t\right)\,d\mathbf{z}_{0:t} \tag{13}$$

### D.2 Posterior predictions

In Figure 2 and Figure 3 we visualize predictions from a filtering posterior. We use the learned inference model to estimate the states of a fixed-length trajectory chunk. Where necessary, we pad the beginning of the observation-action trajectory chunks with zero. Since our inference model yields a smoothing posterior we keep only the very last estimate - a filtering estimate. From this filtering estimate we obtain predictions via ancestral sampling.

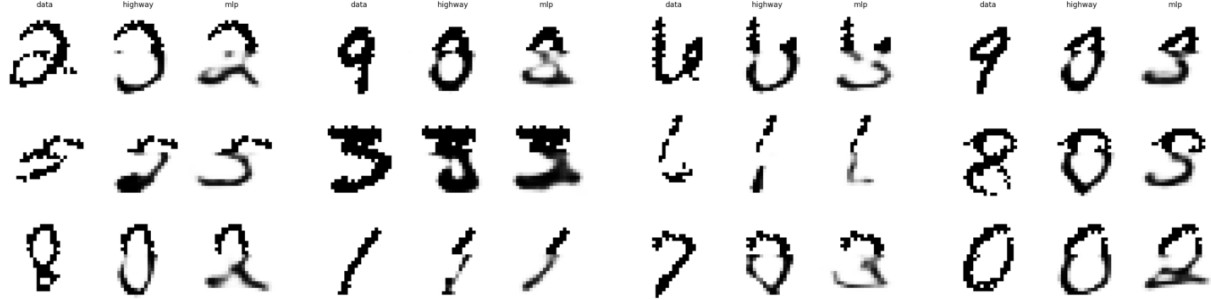

Figure 5: The plots depict predictive posteriors with a prefix length of 14 time steps for ten example trajectories from the validation dataset. The total trajectory length is 28 time steps.

### D.3 Additional visualizations for performance comparison

Figure 6 displays the same results as in Figure 1, however, we choose a different visualization. We observe that a majority of training experiments falls below the line of equivalent performance, hence, for a given hyperparameter set the highway transition model tends to be better. In Figure 6(b) and Figure 6(c), we observe that the models with the smallest validation loss, i.e. dots in the bottom-right, fall below the lines of equivalent performance. We can conclude that the best models are those with highway transitions. Naturally, the conclusions are limited to these datasets and generally only indicative.

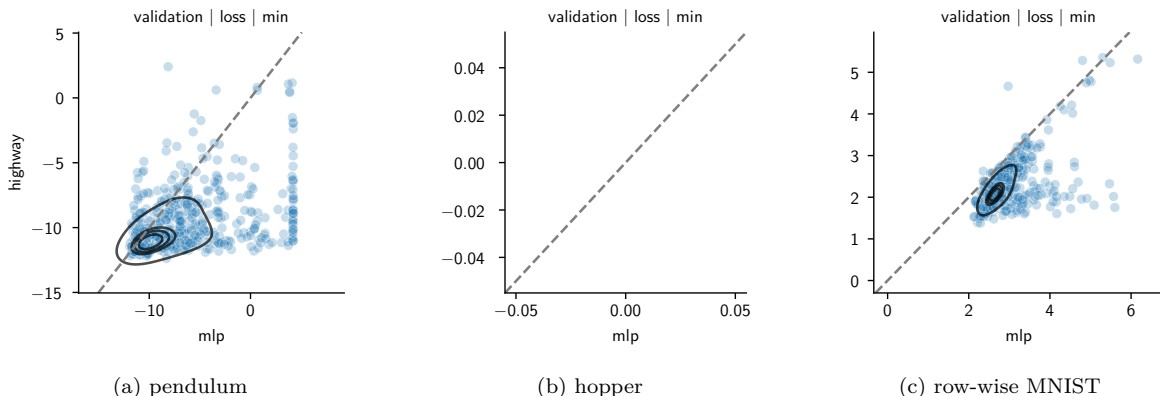

(a) pendulum        (b) hopper        (c) row-wise MNIST

Figure 6: The plots scatter parameter-aligned training runs of highway and mlp transitions against one another. Again, *min* refers to the minimum value throughout training. The dashed diagonal line demarks the line of equivalent performance of both transition variants. The contour lines are estimates of the 0.5, 0.85, 0.9, and 0.95 percentiles.

### D.4 Rendering of hopper observation trajectories

**Input:** Observation sequence array
**Output:** Rendered images
env = make_env("Hopper-v2") ;                 // Initialize Hopper environment
Initialize rendered_images as empty list;
**foreach** *observation in sequence* **do**
    **if** *observation length == 12* **then**
        qpos ← observation[0:5] ;            // First 6 positions including rootx
        qvel ← observation[6:11] ;                // Last 6 velocity values
    **end**
    **else**
        qpos ← [0] ;            // Initialize with rootx position as zero
        qpos[1:5] ← observation[0:4] ;          // Fill remaining positions
        qvel ← observation[5:10] ;          // Extract velocity components
    **end**
    env.set_state(qpos, qvel) ;           // Update environment state
    img = env.render() ;           // Capture rendered frame
    Append img to rendered_images
**end**
Cleanup environment resources;
**return** *rendered images*;
**Algorithm 1:** For rendering hopper observation trajectories, we manually convert environment observations back to the physical system state. Note that, we make unintended use of the D4RL code and, hence cannot test properly.

