# OpenReview forum: "Doubly residual transitions for deep variational state-space models"
_TMLR — Rejected by TMLR_

### Review · Reviewer_nwBP · 2025-08-07

**Summary Of Contributions:**

This paper proposes incorporating highway connections into the deterministic component of variational state-space models and evaluates this modification against standard fully-connected MLP baselines. The authors demonstrate performance improvements across three datasets.

The paper's primary strength is its clarity of presentation. The methodology is clearly explained and the writing is accessible, making the approach readily understandable.

The main limitation of this work is the narrow scope of architectural exploration. While the authors demonstrate that highway connections improve performance, they do not provide comparative analysis with other architectural modifications that could yield similar benefits. Specifically, the paper would benefit from ablation studies comparing:

* Residual connections
* Different gating mechanics (e.g., separate vs. shared parameters for transform and carry gates)
* Alternative skip connection variants

Without these comparisons, it remains unclear whether the observed improvements are specific to highway connections or represent a more general benefit from enhanced gradient flow and representation learning in the deterministic component. This limits the paper's contribution to understanding which architectural principles drive the performance gains.

**Additional Comments:**

Clarity issue in Section 3.1: The phrasing in the first two sentences is unclear, particularly the reference to "the former." It is ambiguous which previously mentioned concept this refers to, making the text difficult to follow. Please revise these sentences to explicitly state what is being referenced rather than using pronouns like "the former" and "the latter."

**Audience:**

Yes

**Audience Explanation:**

Parts of the TMLR audience are interested in state-space models

**Broader Impact Concerns:**

The reviewer sees no reasons to be concerned.

**Claims And Evidence:**

No

**Claims Explanation:**

"The evaluation methodology weakens the strength of evidence by reporting only validation performance without test set results or special hold-out procedures, raising concerns about potential overfitting to the validation split.

**Requested Changes:**

1. Evaluation methodology: Report performance on a proper test set rather than validation data to demonstrate generalization and avoid potential overfitting concerns. The current validation-only evaluation weakens the evidence for the claimed improvements.
2. Figure clarity: Provide explanation for the apparent visual discrepancy in Figure 2, where the MLP baseline appears to show better trajectory reconstruction for the pendulum task despite reported inferior performance metrics.
3. Ablation studies: Conduct comparative experiments with other architectural modifications (e.g., standard residual connections, different gating mechanisms, skip connections) to determine whether the observed improvements are specific to highway connections or reflect a broader benefit from enhanced information flow architectures and adjust claims accordingly. This would strengthen the claims about the specific value of highway connections versus general shortcut/skip connection mechanisms.

---

### Review · Reviewer_qke5 · 2025-08-19

**Summary Of Contributions:**

In this paper, the authors explore the use of a new deep architecture for transition functions of state-space models, namely highway layers. Then, they propose to include this new transition function architecture in a VAE to learn the underlying distribution of the observations. Finally, the whole proposed procedure is named Variational State-Space Models (VRSSMs).

**Additional Comments:**

None

**Audience:**

Yes

**Audience Explanation:**

The new deep architecture for transition function of SSR and the coupling with VAE make the contribution relevent for TMLR in my opinion.

**Claims And Evidence:**

Yes

**Claims Explanation:**

Globally yes, but some choices and tools have to be more clearly detailed (as explained in the Requested Changes section)

**Requested Changes:**

I have some remarks on this paper.

**Minors**:
1. In the abstract, I suggest further developing the contribution itself even if it means reducing the second part.
2. At the end of the introduction, I suggest to add the outline of the paper to add structure.
3. In Equation (3), wouldn't the lower bound of the second product operator be t=0?
4. In Section 2.2, the explanations. I don't really understand what $g_\theta$ and $h_\theta$ are. Are they neural networks? A gating function like in Equation (4)? Maybe a little example or a figure of the layer would help to better understand.
5. In Equation (7), a $\log$ is missing at the beginning.
6. In Equation (8) and the bullet points, some $x$ and $z$ are not in bold character.
7. In Section 4, I would add a figure showing the evolution of ELBO during the training for the best runs.
8. On Figures 2 and 3, it is not clear that highways outperform MLP. Can you explain this point?
9. I would add some perspectives to this work at the end of the conclusion.

**Majors**:
1. Generally speaking, the litterature review is not really clear. I didn't understand while reading what already exists or not, in particular existing (or not) modelisations of transition functions (deep architecture or not) and/or VRSSMs, even if it is mentionned in Section 5.
2. The contribution section (Section 3) is too small in my opinion and lack of clarity.
- Indeed, you present in Equation (9) a certain transition function. Then, the first paragraph of Section 3.1 brings confusion to the actual use of Equation (9). Does the second part ignored? Is the transition only deterministic? Some expressions are really ambiguous, like "second residual layer" or "identity propagation" for example.
- In Section 3.2, I would add explanations ont the construction of the new ELBO function, namely the calculus. Moreover, I would make appear the parameters to optimise, in particular $\theta_{det}$ and $\theta_{res}$ if so.
- At last, as before, I would maybe add a figure of the proposed architecture to add clarity.
3. In the same way as before, I would precise the effect of the choice made in the first paragraph of Section 4 on the architecture of the model. A more precise description of the comparative method would also add strength to the method and the results.

---

### Review · Reviewer_cZL9 · 2025-08-24

**Summary Of Contributions:**

This paper investigates the use of highway layers as transition functions in deep variational state-space models (VRSSMs). The authors argue that gating mechanisms can improve robustness and performance compared to standard MLP-based transitions. They evaluate their approach on three datasets (Pendulum, Hopper, Sequential MNIST), reporting modest but consistent improvements.

**Additional Comments:**

None

**Audience:**

Yes

**Audience Explanation:**

Yes. While the contribution is incremental, the findings would still interest part of the TMLR audience working on sequential modeling and variational state-space models. In particular, practitioners may appreciate the empirical evidence that highway transitions can improve robustness and ease of tuning compared to standard MLP transitions, making the paper useful and interesting.

**Claims And Evidence:**

No

**Claims Explanation:**

The evidence provided is clear but not entirely convincing. The empirical results do show that highway transitions outperform MLP transitions across several datasets, but the gains are often modest (and visually not clear) and confined to relatively simple benchmarks. Moreover, the comparisons are limited in scope, as the paper does not evaluate against stronger baselines such as GRU-based, S4, or Mamba mechanisms. While the experimental methodology is sound and well-documented, the absence of deeper theoretical analysis or broader empirical validation makes the claims feel somewhat overstated.

**Requested Changes:**

Please see the answer about the claims.

---

### Decision · Action_Editor_BDvp · 2025-10-09

**Recommendation:** Reject

**Audience:**

Yes

**Audience Explanation:**

As illustrated by the reviewers, I believe that at least some individuals in TMLR's audience would be interested in knowing the findings of this paper if the authors had addressed the authors’ concerns. This would allow the reader to correctly understand the scope and contributions of this paper.

**Claims And Evidence:**

No

**Claims Explanation:**

While the authors provide some support to their claim, the submission fails to convince the reviewers. A major issue is the lack of clarity when contextualizing the contribution of this work (see review qke5): the set of benchmarks is very limited (see review cZL9), the experimental process is unclear (e.g. evaluation on validation set – see review nwBP), the particular choice of SSM (and Markov constraint) as a particular setting and how it might proscribe the use of certain architectures as a baseline has not been explained (see review nwBP and cZL9). Additionally, each reviewer has provided a set of requested changes to address those issues, some requiring little labor from the authors. None of them have been acknowledged at all.

**Resubmission Of Major Revision:**

The authors may consider submitting a major revision at a later time.